# Reproducibility report:
# Hate Speech Detection based on Sentiment Knowledge Sharing

1 ## Reproducibility Summary

2 *This report summarises our efforts to reproduce the results presented in the ACL2021 paper Hate Speech Detection*
3 *based on Sentiment Knowledge Sharing by Zhou et al. (2021), as part of the ML Reproducibility Challenge 2021.*

4 **Scope of Reproducibility**

5 The main goal of this reproducibility attempt is to confirm the effectiveness of the hate speech detection framework
6 proposed by Zhou et al. (2021). In particular, our efforts are directed at validating their main claim that sentiment
7 knowledge sharing in a multi-task learning setup improves the performance of the model in predicting hate speech.
8 Besides reproducing their main results, we perform repeated experiments to assess the variability of the scores and
9 perform a hyperparameter search.

10 **Methodology**

11 The authors provide a code-base which is available at `https://github.com/1783696285/SKS`. We reuse the
12 available code, modifying it where necessary and integrating it with a few additional scripts for statistics computation and
13 data preparation. Our code, data and results are available at `https://anonymous.4open.science/r/repro-SKS-A`.

14 **Results**

15 Our findings diverge substantially from the results reported in the original paper. In particular, in our reproduction
16 experiments, including sentiment features hurts the performance of the model in the hate speech detection task
17 (approximately 0.5 to 2.0 F1-score).

18 **What was easy**

19 The paper provides some broad indications with respect to the training details and the code-base is publicly available.
20 Similarly, the data-sets are also freely available and the authors provide links to them in their repository.

21 **What was difficult**

22 The code-base is rather convoluted. Following the instructions included in the authors' repository resulted in a number
23 of exceptions caused by formatting issues, missing code snippets and hard-coded values. Additionally, the lack of a
24 clear and comprehensive documentation contributed to an arduous code review and reproducibility effort.

25 **Communication with original authors**

26 We managed to reach one of the authors and exchange a few messages over GitHub. However, despite multiple attempts,
27 we did not manage to reach the authors per email and get an answer to our questions concerning some aspects of the
28 implementation.

# 1 Introduction

Being able to quickly and reliably detect hate speech in an automatic manner is an important task. Due to the growing number of regulations concerning the use of hate speech and other forms of offensive language online, this topic has gained increasing interest both in academia and industry (Davidson et al., 2017; Schmidt and Wiegand, 2017; Basile et al., 2019; Yin and Zubiaga, 2021).

As in any supervised learning task, the availability and the size of labelled data-sets pose significant challenges. The task is made even more arduous by its multilingual and multi-domain nature. One way to alleviate such problems is to make use of additional data-sets from other, related tasks.

The study by Zhou et al. (2021) that we attempt to reproduce describes a multi-task learning framework for online hate speech detection that relies on the purportedly strong negative sentiment characterising this threatening form of communication. The model presented in the original paper, Sentiment Knowledge Sharing (`SKS`), is a multi-head attention network that predicts whether the input text contains hate speech or not. The main claim of the paper revolves around the fact that the model is (optionally) trained in a multi-task setting for sentiment analysis, and it incorporates information from a dictionary of derogatory words through 'category embeddings' (see Section 3.1 for further details).

Based on experiments carried out on two benchmark data-sets, Zhou et al. (2021) claim that training a model relying both on sentiment information and category embeddings would improve its performance in the task of hate speech detection.

# 2 Scope of reproducibility

The work of Zhou et al. (2021) is based on the intuition that hate speech detection and sentiment analysis are two highly correlated tasks and that hate speech is likely to arise from derogatory words. Our reproducibility attempt tries to verify the following claims:

- A model relying both on Sentiment Knowledge Sharing (`SKS`) and a dictionary of derogatory words scores better than several strong baselines where sentiment features are not considered.

- Ablating the sentiment knowledge component (`-s`) results in a poorer performance, as the model would rely solely on derogatory words features which, despite being likely indicators of hate speech, can make the model too sensitive too false positives (e.g. *I'm so fucking ready!*).

- A model where both sentiment knowledge and derogatory word features are ablated (`-sc`) scores the worst performance.

Besides trying to reproduce the original results (see Table 3 in Zhou et al. (2021)), we perform a hyperparameter search to validate the values reported by the authors. Every experiment we perform is ran multiple times to check whether any observed differences stand when the variability of the scores are taken into consideration.

# 3 Methodology

## 3.1 Model descriptions

The `SKS` model relies heavily on the Mixture-of-Experts layer (MoE) as introduced by Shazeer et al. (2017) and the Multi-gate Mixture-of-Experts (MMoE) model presented by Ma et al. (2018). Its overall architecture consists mainly of three macro-components: an input layer, a sentiment knowledge sharing layer and a gated attention layer.

### 3.1.1 The input layer

In the input layer, word embeddings are used to encode words of each target sentence. Specifically, every token $w_i$ of a given sentence $S = \{w_1, w_2, ..., w_i, ..., w_N\}$ is transformed into a real-valued vector $x_i \in \mathbb{R}^d$. Additionally, given that derogatory words represent a helpful marker of hate speech, each vector $x_i$ is concatenated with a category embedding vector $c_i \in \mathbb{R}^{d^i}$, such that $x_i^{'} = x_i \oplus c_i$.

70 Category embeddings are created on the basis of a dictionary of derogatory words which is used to classify sentences
71 into two categories, either containing derogatory words or not. The result of the classification is encoded as a vector $c_i$
72 and appended to each word embedding $x_i$, such that the encoded sentence is $S = \{x_1', x_2', ..., x_N'\}$.

### 3.1.2 The sentiment knowledge sharing layer

74 The sentiment knowledge sharing component relies on a multi-task learning strategy which, according to the authors,
75 would allow to take advantage of the high correlation between the two tasks of sentiment analysis and hate speech
76 detection. In the proposed implementation, the two tasks share a bottom hidden layer implemented following the
77 Mixture-of-Experts framework (MoE). The MoE layer is made up of multiple identical feature extraction units each of
78 which, in turn, is composed of a multi-head attention layer using 4 heads and two feed forward neural networks.

79 Each unit relies on the idea of multi-head attention introduced by Vaswani et al. (2017), where the input matrix X is
80 mapped to query $Q \in \mathbb{R}^{(n_1 \times d_1)}$, key $K \in \mathbb{R}^{(n_1 \times d_1)}$, and value $V \in \mathbb{R}^{(n_1 \times d_1)}$ using linear transformations. Given these
81 three matrices the attention parameters are computed as follows:

$$\text{Attention}(Q, K, V) = \text{softmax}\left(\frac{QK^\top}{d_1}\right) V \tag{1}$$

82 In the implementation proposed by Zhou et al. (2021) $K = V$ and $d_1$ corresponds to the number of hidden layer units.
83 The $ith$ output of the multi-head attention mechanism is:

$$M_i = \text{Attention}(QW_i^Q, KW_i^K, VW_i^V) \tag{2}$$

84 where the parameter matrix $W_i^Q \in \mathbb{R}^{n_1 \times \frac{d_1}{l}}$, $W_i^K \in \mathbb{R}^{n_1 \times \frac{d_1}{l}}$ and $W_i^V \in \mathbb{R}^{n_1 \times \frac{d_1}{l}}$. All outputs are then concatenated
85 and multiplied by $W^O$ to obtain the final feature representation $H^s = \text{concat}(M_1, M_2, ..., M_l)W^O$.

86 Finally, the authors decide to use both maximum and average pooling (Shen et al., 2018) to fuse the feature representa-
87 tions, concatenating the two results:

$$P_m = \text{Pooling\_max}(H^s) \tag{3}$$

$$P_a = \text{Pooling\_average}(H^s) \tag{4}$$

$$P_s = \text{concat}(P_m, P_a) \tag{5}$$

### 3.1.3 The gated attention layer

89 The third macro-component is a gated attention mechanism which allows to select a subset of the feature extraction
90 units from the previous layer. The output $g^k(x)$ of a specific gate $k$ corresponds to the probability of selecting a specific
91 unit. The subset of units selected through this process are then weighted and summed to get the final representation
92 $f^k(x)$ of a given sentence, which is passed to a feed-forward neural network to detect hate speech:

$$g^k(x) = \text{softmax}(W_{gn} * \text{gate}(x)) \tag{6}$$

$$f^k(x) = \sum_{i=1}^{n} g^k(x)_i f_i(x) \tag{7}$$

$$y_k = h^k f^k(x) \tag{8}$$

## 3.2 Datasets

Following in the steps of Zhou et al. (2021), we test the model and report results on two public hate speech data-sets: SemEval2019 Task-5 (SE, Basile et al., 2019)[1] and Davidson (DV, Davidson et al., 2017).[2]

The SE data-set contains a total of $13,000$ tweets and is divided into training-, validation- and test-set, consisting of $9,000$, $1,000$ and $3,000$ samples, respectively. The training-set contains $3,783$ instances of hate speech and $5,217$ instances that are not. In the validation-set $427$ samples are classified as hate speech and $573$ as non-hate speech. The test-set is split into $1,260$ hate speech samples and $1,740$ non-hate speech ones.

The DV data-set contains a total of $24,783$ manually labelled tweets. Each tweet is assigned to either one of three classes: hate speech ($1,430$), offensive language ($19,190$) or neither ($4,163$). Zhou et al. (2021) merge the last two classes together and obtain $1,430$ tweets classified as hate speech and $23,353$ classified as non-hate speech.

Finally, the model relies also on a sentiment data-set obtained from Kaggle.[3] The original authors only use the training-set which contains $31,962$ tweets, $2,242$ of which are classified as having a negative sentiment, while the remaining $29,720$ a positive one.

## 3.3 Hyperparameters

We begin our reproducibility attempt, relying solely on the hyperparameters reported in the original paper. Our results are summarised in Table 1.

In the input layer, all word vectors are initialised using Glove Common Crawl Embeddings (840B Token) with a dimension of $300$, while category embeddings are randomly initialised and have a dimension of $100$.

In the sentiment knowledge sharing layer, the multi-head attention mechanism is implemented using $4$ heads. The two feed-forward networks in each expert unit have one layer with $400$ units and two layers with $150$ units, respectively. However, contrary to what we see in the implementation, it is worth noting that the original paper reports $200$ units for the second network. After each layer a dropout rate of $0.1$ is used.

The model is trained by mini-batches of $512$ instances for $15$ epochs, using the RMSprop optimiser and a learning rate of $0.001$. The original authors report using learning rate decay and early stopping to avoid overfitting.

### 3.3.1 Hyperparameters tuning

The original work does not provide any details regarding hyperparameters tuning and upon contacting the authors to inquire about it we received no answers. Thus, we attempt to tune learning rate ($10^{-6}$ to $10^{-1}$, on a log scale), batch size (from 32 to 1024, on a $\log_2$ scale) and dropout rate ($0.0$ to $0.4$ with increments of $0.1$) on the SE data-set using grid-search with 60 epochs and find that the respective optimal values are $0.001$, $256$ and $0.0$. However, the values indicated in the original paper performs similarly. Considering the model variation (see Table 1 and Figure 1), the differences can easily be attributed to the model variance, due to random initialization.

## 3.4 Experimental setup and code

We try to reproduce the results presented in Table 3 of the original paper (Zhou et al., 2021). For both data-sets the authors train three models: `SKS`, which relies both on sentiment knowledge sharing and category embeddings; `-s`, a model where the sentiment knowledge sharing component is ablated; `-sc`, a model that does without both sentiment knowledge sharing and category embeddings. We rely largely on the Tensorflow implementation (Abadi et al., 2015) made available by the authors, modifying it where necessary and integrating it with a few additional scripts for statistics computation and data preparation.

For each result reported in the original paper we repeat the corresponding experiment 10 times. Specifically, for each repetition the model is reinitialised and trained over 15 epochs. We keep the results from the best epoch of each repetition and then compute the average and the standard deviation for the originally employed measures i.e. accuracy and macro-F1 score for the SE data-set and accuracy and weighted-F1 score for the DV data-set.

---

[1] http://hatespeech.di.unito.it/hateval.html
[2] https://github.com/t-davidson/hate-speech-and-offensive-language/tree/master/data
[3] https://www.kaggle.com/dv1453/twitter-sentiment-analysis-analytics-vidya

Given that the DV data-set is highly unbalanced, the authors use a 5-fold cross-validation approach to measure the performance of each model. We follow the same approach and adopt the 10 times repetition strategy for each fold.

Our code, data as well as the final and intermediate per-iteration results are available at https://anonymous.4open.science/r/repro-SKS-A.

### 3.5 Computational requirements

To run our experiments we use a NVIDIA TITAN Xp with a 12 GB memory. Training the models on the SE data-set took approximately 24 minutes for the `SKS` model and 7 minutes both for the `-sc` and `-s` model. On the DV data-set the training took approximately 3 hours for the `SKS` model and 2 hours both for the `-sc` and `-s` model. The hyperparameters tuning step on the SE data-set took approximately 33 hours.

## 4 Results

We report the original results along the ones we obtained using the specified hyperparameters in Table 1. Comparing our findings with those reported by the original authors we observe a discrepancy in all three measures, Accuracy, macro-F1 and weighted-F1 score, for both data-sets. In the SE data-set, the most notable differences concern the results of the `SKS` and `-s` models. In the DV data-set, there are some noteworthy discrepancies only with respect to the `SKS` model.

Looking at the mean scores we obtain on the SE data-set, the `SKS` model does not outperform both ablated versions `-s` and `-sc`, thus contradicting the first and second claim in Section 2. In fact, while `SKS` obtains an Accuracy of 61.04 and a macro-F1 score of 60.88, the `-s` model outperforms it, reaching an Accuracy and a macro-F1 score of 64.17 and 63.05, respectively. On the other hand, the third claim appears to hold. With an Accuracy of 60.52 and a macro-F1 score of 60.47 the `-sc` model is the one registering the worst performance.

Turning to the DV data-set, none of the claims appear to be substantiated by our findings either. The `SKS` models scores the lowest with an Accuracy of 93.63 and a weighted-F1 score of 93.62, while the ablated versions `-s` and `-sc` register similar values for both metrics, with an Accuracy of 93.99 and 93.98 and a weighted-F1 score of 94.11 and 94.12, respectively.

| Model | DV | | | | SE | | | |
| | Acc | | F1 (weighted) | | Acc | | F1 (macro) | |
| | Orig. | Repro. | Orig. | Repro. | Orig. | Repro. | Orig. | Repro. |
|---|---|---|---|---|---|---|---|---|
| `-sc` | 94.0 | 93.98 ($\pm$1.61) | 94.0 | 94.12 ($\pm$1.73) | 59.6 | 60.52 ($\pm$1.44) | 59.3 | 60.47 ($\pm$1.40) |
| `-s` | 94.5 | 93.99 ($\pm$1.49) | 94.3 | 94.11 ($\pm$1.58) | 61.3 | 64.17 ($\pm$0.99) | 61.3 | 63.05 ($\pm$0.63) |
| `SKS` | 95.1 | 93.63 ($\pm$2.09) | 96.3 | 93.62 ($\pm$2.37) | 65.9 | 61.04 ($\pm$1.81) | 65.2 | 60.88 ($\pm$1.64) |

Table 1: For each data-set and performance measure we report each model's original (Orig) results on the left and the reproduced (Repro) ones on the right, including the standard deviation of the reproduced score.

For a visual inspection of the results presented in Table 1 we also plot box plots of the scores obtained in multiple reproduction attempts in Figure 1. Despite some overlap in the range of the obtained scores, the median scores of the `SKS` model is lower than those of the ablated versions. The figure also shows that the scores reported in the original paper fall within the range $\pm 1.5$ standard deviation from the mean of the scores of the multiple reproduction experiments. However, for both data-sets, the original scores of `SKS` is substantially above this range.

### 4.1 Alternative metrics

The original paper reports macro- or weighted-averaged F1 scores, with the motivation of comparability to earlier research on these data-sets. However, the task at hand is a binary classification task with a clear positive class. Incorporating the negative class score through averaging does not allow to assess the success of the classifier on the task. Furthermore, relying on weighted averaging without having a justified set of weights, but using weights proportional to the support of each class, rewards classifiers with majority bias even further.

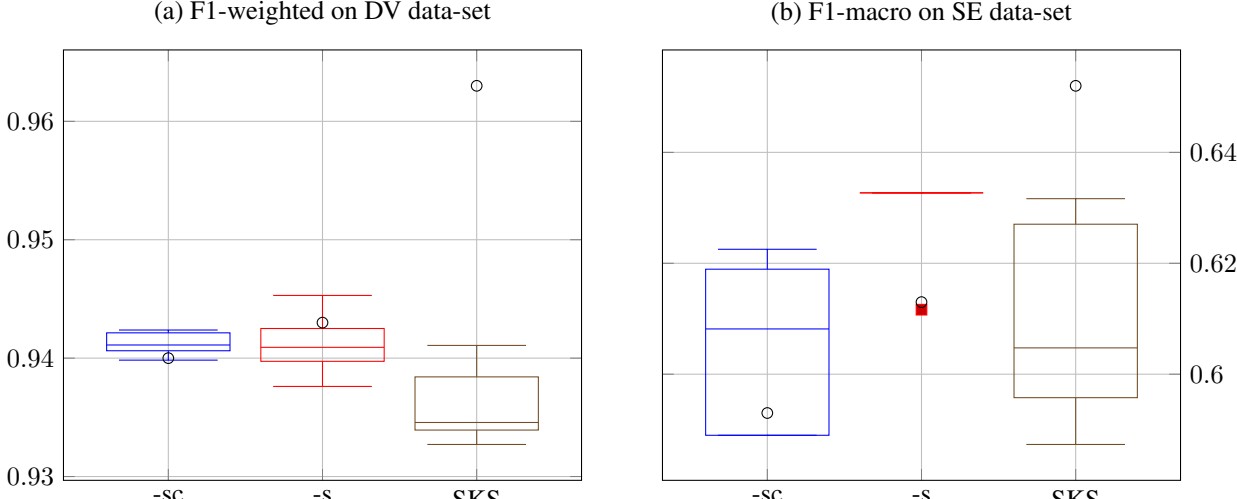

Figure 1: Box plots of (a) F1-weighted on the DV data-set and (b) F1-macro on the SE data-set, from repeated experiments with different initialisations. Circles represent the scores reported in the original article. Red square in (b) indicates the single outlier for the `-s` option on this data-sets. The rest of the scores are equal to the median. Note that the y-axes do not have the same scale.

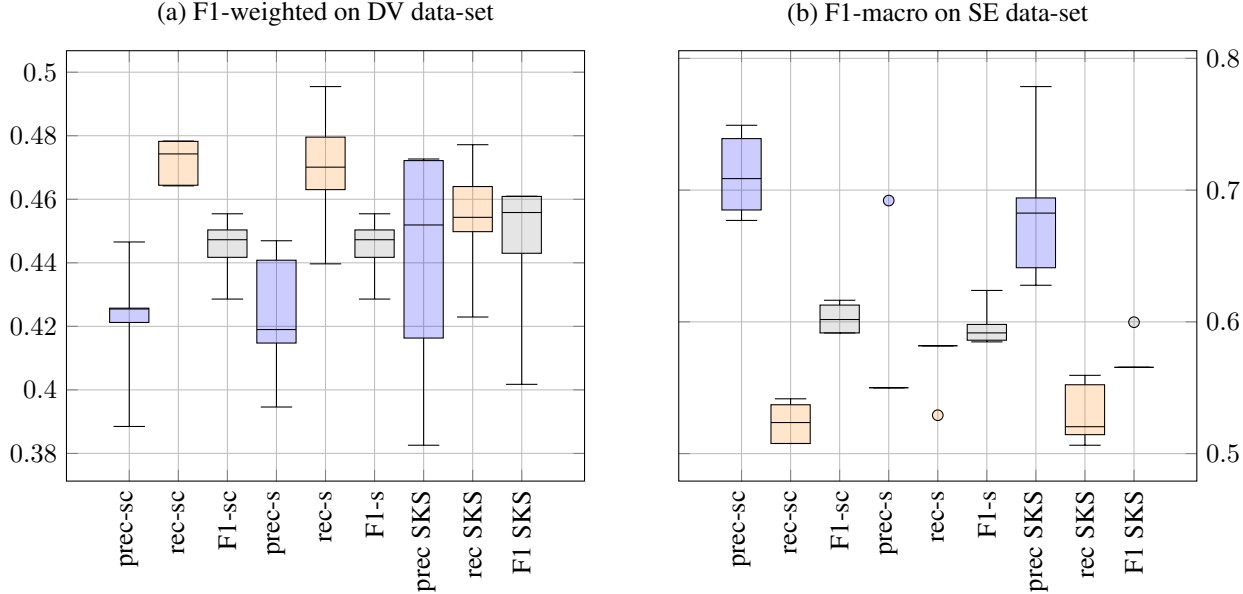

Figure 2: Box plots of binary precision (blue), recall (orange), and F1-scores on (a) the DV data-set and (b) the SE data-set, from repeated experiments with different initialisations. Note that the y-axes do not have the same scale.

To present a more interpretable impression of the success of each model and provide further insight into the differences based on model ablation and alternations, Figure 2 depicts the distribution of precision and recall for the different reproduction experiments carried out on the two data-sets.

The plots indicate that, despite a large overlap, jointly learning sentiment analysis (SKS) improves the precision of the hate speech detection on the DV data-set. Despite having a negative impact on the recall, this also yields a slightly better median F1 score. The effect of the sentiment task is mostly negative on the SE data.

# 5   Discussion

The reproducibility results from Section 4 do not support the claims outlined in Section 2 for either data-sets. In particular, our findings seem to suggest that the multi-task learning approach implemented by the authors to allow the SKS model to extract sentiment features and apply them to hate speech detection does not yield the expected results. However, considering the lack of a comprehensive documentation, the convoluted structure of the code-base and the insufficient communication with the original authors it is hard to draw definitive conclusions. In fact, there are a number of plausible explanations as to why our findings diverge from those reported in the original paper.

For instance, considering the slight difference between the optimal hyperparameters we found and those reported by Zhou et al. (2021), and large variation of the model scores, one could speculate that, at least for part of the experiments, the authors employed some parameters which have not been reported. This would also explain the difference between some of the values indicated in the paper those used in the provided implementation.

Another explanation could lie in the fact that we inadvertently deviated from the original implementation while trying to fix some of the issues we faced in running the code-base. Whenever information was missing or not completely clear assumptions had to be made, and we tried to approximate the original results by trial and error. This was the case for the `-sc` model where the procedure to ablate the category embeddings component was not given and the answer we received from the authors did not help us overcome the problem.

The main intuition behind the original study is the fact that hate speech typically carries a negative sentiment. Hence, this relation between the tasks would help the model to identify hate speech better (arguably by increasing recall). However, a manual inspection of the data-set, on the other hand, suggests that it may actually be surprising for a classifier informed by sentiment analysis to help hate speech detection. Both data-sets are collected using keywords that are likely to contain hate speech, and negative classes constitute posts that are either offensive (but not hate speech), or people counteracting earlier offensive content. Hence, the sentiment on the negative class is not necessarily positive and helpful for discriminating hate speech *in these data-sets*. However, in a more realistic environment, the author's proposal may be correct. Given more 'normal' negative class instances, learning sentiment analysis jointly is likely to inform the hate speech detection.[4] The binary evaluation metrics presented in Figure 2 indicates that at least on the DV data-set, the addition of sentiment may have some positive effects. Understanding the reasons for these differences, and improving the joint learning model is a possible direction for the future research.

## 5.1   What was easy

The paper provides some broad indications with respect to the training details and both the data-sets and the code-base are open-sourced.

## 5.2   What was difficult

The lack of a comprehensive documentation, the convoluted structure of the code-base and the insufficient communication with the original authors contributed to an arduous code review and reproducibility effort.

## 5.3   Communication with original authors

We first tried to review and run the provided code-base by ourselves. However, after stumbling on a number of issues related to how the data-sets were being processed and how to run the `-sc` model ablating category embeddings, we decided to reach out to the authors through GitHub. One of the corresponding authors provided some indications which, unfortunately, did not help us overcome the problems at hand.

We also tried to contact the authors per email twice, inquiring about some aspects of the model implementation as well as the procedure they followed to tune the hyper-parameters. However, we never received an answer.

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
