# OpenReview forum: "Reproducibility report: Hate Speech Detection based on Sentiment Knowledge Sharing"
_ML_Reproducibility_Challenge/2021/Fall — RC2021_

### Official Review · Reviewer_ygKr · 2022-03-01
**Reviewing "Reproducibility report: Hate Speech Detection based on Sentiment Knowledge Sharing"**

**Rating:** 6
**Confidence:** 3

**Review:**

The paper addresses an important and timely topic of tooling hate speech detection on social media. It studies reproducing a 2021 ACL study by Zhou et al. (2021), building on its report, codebase, and communication with the original authors. In addition to trying to verify the three key claims made by Zhou et al (2021), as well as to reproduce their experimental evaluation results, this paper explores the hypotheses of hate speech detection and sentiment analysis being highly correlated tasks with the former arising from derogatory wordings that was underpinning the original study. This careful research orientation is exceptional in a reproducibility report and well articulated in the report. However, in my opinion, the authors should be careful in not overstating their own claims; perhaps they wish to tone down their own contributions and critique given to Zhou et al. (2021).

Specific comments

Please consider communicating the key message of the paper already earlier as now it took until the Discussion section in the very end to realise that the outcomes of the reproducibility report were actually contrary to the original findings.

Please try to separate methods from results in Section 4 more carefully.

Please consider critiquing Zhou et al. (2021) in a way that is more collegial.

Noting that existing datasets were use, please address data access and usage rights briefly in the paper. For ethical code of conduct, commenting on this aspect is important. This is to assure the readers that the proper approvals and permissions were considered and followed what comes to assuring that these datasets were originally collected in an ethical manner and their use was appropriate for the purposes of this study.

Discussing broader impacts of this study is also recommended.

Please add a license to your code release (e.g., MIT or GNU versions might be applicable here).

Please punctuate math.

---

### Official Review · Reviewer_FSqq · 2022-03-08
**Excellent report and well organized code repo**

**Rating:** 9
**Confidence:** 3

**Review:**

Both the submitted report and the code are well organized and easy to follow.

The author conducted very detailed analysis on the state of reproducibility and did extensive hyperparameter search. They also provided reasonable alternative explanations on why their results diverting from the original paper.

---

### Meta-Review · Program_Chairs · 2022-04-07

**Recommendation:** Accept
**Confidence:** 4

**Metareview:**

An interesting contribution to ML reproducibility with some minor improvements that can be made -- see reviewer ygKr's points, notably in terms of separating methods and results, and putting salient points from the Discussion earlier on in the paper (e.g. in the Introduction) to make its point and findings clearer.

---

### Decision · Program_Chairs · 2022-04-09

**Decision:**

Accept

**Comment:**

Following the recommendation of reviewers and meta-reviewer, the paper is accepted for ML Reproducibility Challenge 2021, and will be published in the upcoming special edition of ReScience Journal.